# Induction of Acquired Resistance towards EGFR Inhibitor Gefitinib in a Patient-Derived Xenograft Model of Non-Small Cell Lung Cancer and Subsequent Molecular Characterization

**DOI:** 10.3390/cells8070740

**Published:** 2019-07-18

**Authors:** Julia Schueler, Cordula Tschuch, Kerstin Klingner, Daniel Bug, Anne-Lise Peille, Leanne de Koning, Eva Oswald, Hagen Klett, Wolfgang Sommergruber

**Affiliations:** 1Charles River Discovery Research Services Germany GmbH, Am Flughafen 12-14, 79108 Freiburg, Germany; 2Lehrstuhl für Bildverarbeitung, RWTH-Aachen University, 52074 Aachen, Germany; 3Institut Curie, PSL Research University, Department of Translational Research, 75005 Paris, France; 4University of Applied Sciences, FH Campus Wien, 1200 Vienna, Austria

**Keywords:** NSCLC, acquired resistance, EGFR inhibition, PDX, reverse phase protein array, whole exome sequencing

## Abstract

In up to 30% of non-small cell lung cancer (NSCLC) patients, the oncogenic driver of tumor growth is a constitutively activated epidermal growth factor receptor (EGFR). Although these patients gain great benefit from treatment with EGFR tyrosine kinase inhibitors, the development of resistance is inevitable. To model the emergence of drug resistance, an EGFR-driven, patient-derived xenograft (PDX) NSCLC model was treated continuously with Gefitinib in vivo. Over a period of more than three months, three separate clones developed and were subsequently analyzed: Whole exome sequencing and reverse phase protein arrays (RPPAs) were performed to identify the mechanism of resistance. In total, 13 genes were identified, which were mutated in all three resistant lines. Amongst them the mutations in NOMO2, ARHGEF5 and SMTNL2 were predicted as deleterious. The 53 mutated genes specific for at least two of the resistant lines were mainly involved in cell cycle activities or the Fanconi anemia pathway. On a protein level, total EGFR, total Axl, phospho-NFκB, and phospho-Stat1 were upregulated. Stat1, Stat3, MEK1/2, and NFκB displayed enhanced activation in the resistant clones determined by the phosphorylated vs. total protein ratio. In summary, we developed an NSCLC PDX line modelling possible escape mechanism under EGFR treatment. We identified three genes that have not been described before to be involved in an acquired EGFR resistance. Further functional studies are needed to decipher the underlying pathway regulation.

## 1. Introduction

Lung cancer is still the leading cause of cancer related deaths worldwide, accounting for 1.76 million deaths in 2018 (http://www.webcitation.org/77V92AxNh). Non-small cell lung cancer (NSCLC) represents 85% of all lung cancer cases. Besides surgery, chemo-, radio-, and in recent years, immunotherapy-targeted approaches are one of many treatment options. By targeting cancer-specific molecules and signaling pathways, the systemic treatment options display markedly less nonspecific toxicities. In a high proportion (~30%) of non-small cell lung cancer (NSCLC) patients, the oncogenic driver of tumor growth is the constitutively activated epidermal growth factor receptor (EGFR, HER-1/ErbB1), a receptor tyrosine kinase of the ErbB family, which has been identified as an anticancer target [1]. Apart from EGFR-targeting monoclonal antibodies (mAbs), small-molecule tyrosine kinase inhibitors (TKIs) are in clinical use and approved for fist line therapy in NSCLC. In general, tyrosine kinases are central targets as they not only play an important role in the modulation of growth factor signaling in the initial phase of tumor establishment but also during other stages of tumorigenesis, such as escape from treatment or epithelial-mesenchymal transition (EMT), including prognostic power [1,2,3,4,5,6]. Despite the overall good response towards EGFR, TKI’s acquired resistance occurs in most of the patients after 9 to 14 months of therapy [7]. Numerous mechanisms of acquired resistance towards EGFR-TKIs are reported, ranging from secondary mutations in the *EGFR* gene to amplification or overexpression of molecules within the EGFR signaling cascade, like MET or hepatocyte growth factor (HGF) [7,8,9,10]. Beside the well-described T790M mutation, novel mutations in the *EGFR* gene were determined recently. On top of that, non-EGFR related mutations, for example, TP53, were described to be enriched in patients developing an acquired resistance towards EGFR TKIs [11]. To overcome those acquired resistances, preclinical tools to study their development as well as to test new drugs overcoming those mechanisms are urgently needed. In recent times, there have been multiple efforts in the industry as well as academia to establish large panels of well-characterized patient-derived xenograft (PDX) models covering a wide range of different tumor types. Indeed, these collections are becoming the preferred research tool to optimize the drug development process at multiple steps, in particular for target validation, pharmacology, and translational studies [12,13,14,15]. Currently, these collections represent the complexity of tumor heterogeneity and the molecular diversity of human cancers. In our facility, we established a panel of 85 NSCLC PDX models, representing the molecular landscape of NSCLC. In the present study, we derived three Gefitinib-resistant sublines of an NSCLC PDX model (LXFA 677) that was originally sensitive towards EGFR targeted treatment. The PDX model was derived from a patient who received first line Cisplatin therapy and was EGFR as well as KRAS wt, which classified it for second line EGFR treatment. The sublines were established by constant treatment with Gefitinib over a period of a minimum of three months. We characterized the emerging resistant sublines thoroughly on the genetic as well as protein level to decipher the biological difference among them and in comparison, to their parental line. These data led to a better understanding of the evolution of resistance under EGFR TKI treatment. Furthermore, the sublines will serve as research tools to develop next generation compounds, improving the life expectancy of NSCLC patients with acquired resistance.

## 2. Materials and Methods

### 2.1. PDX Establishment

This study was carried out in strict accordance with the recommendations in the Guide for the Care and Use of Laboratory Animals of the Society of Laboratory Animals (GV SOLAS). All animal experiments were approved by the Committee on the Ethics of Animal Experiments of the regional council (Permit Numbers: G-09/58, G-13/13 and G13/43). After written informed consent, tumor tissue from NSCLC patients undergoing surgery was placed in a storage solution and transported within a few hours to Charles River. Incoming material of every donor patient received a chronological unique number, which was subsequently used to identify the corresponding PDX model. To facilitate the overview of the PDX models, each model name starts with a three to four letter code identifying the tumor type. For NSCLC, three different subtypes were defined: LXFA, lung cancer Freiburg adeno carcinoma; LXFE, lung cancer Freiburg epidermoid carcinoma; and LXFL, lung cancer Freiburg large cell carcinoma. Four to six week old female NMRI nu/nu mice (Charles River, Germany) placed under isoflurane anesthesia received tumor implants subcutaneously in both flanks. During the first passages, mice were monitored for tumor growth for up to 12 months. When stable tumor growth could be determined, mice were sacrificed and tumor material was implanted into new recipient mice. In addition, xenograft material was stored in liquid nitrogen for future implantation or fixed in formalin and stored in liquid nitrogen for subsequent analyses. A PDX was defined as established when stable growth over at least three passages and regrowth from cultures stored in liquid nitrogen could be observed. The percentage of tumor implants displaying stable growth (take rate) and passage time were recorded for every model and every individual passage. Tumor growth was determined by a two-dimensional measurement with calipers weekly or biweekly depending on the growth characteristics of the respective PDX model. Tumor volumes were calculated according to the following equation: Tumor Vol (mm^3^) = a (mm) × b^2^ (mm^2^) × 0.5, where “a” is the largest diameter and “b” is the perpendicular diameter of the tumor representing an idealized ellipsoid. Animals had to be sacrificed when tumor volume reached 1800 mm^3^. The NSCLC PDX model, LXFA 677, was derived from a biopsy from a 62 years old male suffering from non-pre-treated adenocarcinoma of the lung. The metadata of the donor patient are listed in Table 1. The tumor growth behavior of the established PDX model LXFA 677 is depicted in Appendix A.

### 2.2. Treatment Experiments In Vivo

Implantation was performed similar to PDX establishment, except that animals received unilateral tumor implants. Animals and tumor implants were monitored daily until the maximum number of implants showed clear signs of beginning solid tumor growth. At randomization, the volume of growing tumors was initially determined. Animals bearing 50 to 250 mm^3^ tumors, preferably 80 to 200 mm^3^, were distributed into experimental groups, with comparable median and mean tumor volumes. The day of randomization was designated as day 0 of an experiment and was also the first day of dosing. Gefitinib (#G-4408-1G, LC Laboratories, USA) was applied once a day via oral gavage in doses ranging from 6.25 to 100 mg/kg/d. Details are depicted in Table 2. The relative volume of an individual tumor on day X (RTVx) was calculated by dividing the absolute volume (mm^3^) of the respective tumor on day X (Tx) by the absolute volume of the same tumor on the day of randomization, i.e., on day 0 (T0), multiplied by 100, as shown by the following equation:RTVx [%] = Tx/T10∗100.(1)

Tumor inhibition on a particular day (T/Cx) was calculated from the median RTV of a test group and the median RTV of a control group multiplied by 100, as shown by the following equation:T/Cx [%] = median RTVx treated group/median RTVx control group∗100.(2)

The minimum T/C (%) value recorded for a particular group during an experiment represented the maximum anti-tumor activity for the respective compound (=optimal T/C). All experiments were performed according to the relevant animal welfare guidelines published by FELASA and GV-SOLAS in an AAALAC accredited animal facility. Tumor volume and body weight were determined twice per week. A flow chart indicates the number of mice and passaging intervals during the generation of the three sublines (Appendix A).

### 2.3. Formalin-Fixed Paraffin-Embedded (FFPE) Samples, Tissue Micro Array (TMA), and Immune-Histochemistry (IHC)

FFPE: Tumors were collected immediately after euthanasia of the donor animal. FFPE fixation was performed in 10% neutral buffered formalin for 24 h followed by routine processing and embedding into paraffin

TMA: Whole tumor sections (4 μm) were cut and stained with Hematoxylin-Eosin (H&E). H&E sections of the xenografts were studied by light microscopy and representative areas marked on the slides. Xenograft biopsies, 1 mm in diameter, were taken from the corresponding area in the paraffin block and arrayed in duplicates into a new recipient block as described.

IHC: After antigen retrieval, 5 μM FFPE tissue sections were incubated with anti-human EGFR Antibody (1:36; Dako Cat# M7239, Lot 20055023) overnight at 4 °C, followed by DAB staining and hematoxylin counterstaining. Image analysis: Digitalized images of the IHC slides were evaluated to determine the percentage of EGFR positive areas using OSANO software. A computerized analysis for digitized whole-slide images of the samples was used to quantify the EGFR expression using color classification and morphological image processing techniques.

### 2.4. RNA Isolation

Total RNA was isolated from the snap frozen tumor xenograft samples with the “mirVana miRNA Isolation kit” (Ambion, Carlsbad, CA, USA) according to the manufacturer’s instructions. The genomic DNA was digested using “RNase-free DNase Set” (Qiagen, Hilden, Germany). The quality and quantity of the RNA preparations were controlled using the Bioanalyzer (Agilent Technologies, Palo Alto, CA, USA) and the NanoDrop 2000 (Thermo Scientific, Waltham, MA, USA).

### 2.5. DNA Isolation

DNA was extracted from snap frozen tumor xenograft samples. Tumors were digested with proteinase K at 55 °C overnight and the lysates were digested with DNase-free RNase (Qiagen). DNA was extracted with phenol:chloroform:isoamylalcohol and precipitated with ethanol. DNA pellets were washed and resuspended in TElow (Tris 10 mM pH8, EDTA 0.1 mM pH8). The integrity of each DNA preparation was checked on a 1.3% agarose gel and the purity analyzed using the NanoDrop 2000 (Thermo Scientific, Waltham, MA, USA).

### 2.6. Reverse Transcription

RNA samples were reverse transcribed in a final volume of 20 µL (200 ng total RNA, 10 mmol/L DTT, 1 µg hexamer primers, 2 U MMLV Reverse Transcriptase (Invitrogen, Carlsbad, CA), 40 U of RNasin (Promega, Madison, WI), 0.5 mmol/L each dNTP (Promega), 1X reaction buffer). After cDNA synthesis, samples were adjusted to a final volume of 200 µL and stored at −20 °C until use. Then, 5 µL cDNA (corresponding to a quantity of 5 ng reverse-transcribed RNA) was used for each PCR.

### 2.7. qRT-PCR

qRT-PCR assays for human c-MET, ERBB2, ERBB3, HGF, PTEN, and AXL were developed and used on DNA and cDNA to quantify the gene copy number and the mRNA expression of these genes in the resistant LXFA 677 models, respectively. A qRT-PCR GAPDH assay and a qRT-PCR 18S ribosomal RNA assay were developed and used to detect these housekeeping genes on DNA and cDNA, respectively. qRT-PCR primers were designed with the online program Primer3, and the primer sequences are indicated in the table below (Table 3).

qRT-PCR was performed in 40 cycles on a StepOnePlus™ system (Applied Biosystems) using 2x SYBR Green I Master Mix (Kapa Biosystems). The c-MET, ERBB3, PTEN, HGF, and AXL data contained on cDNA was normalized using the 18S ribosomal RNA content of the sample. The c-MET and ERBB2 data obtained on DNA was normalized using the 18S ribosomal DNA content of the sample. The gene copy number and expression results are expressed in arbitrary units (AU) and were calculated as described below:Gene of interest expression or copy number = 2 (Ct_housekeeping gene_ − Ct_gene of interest_).(3)

If no amplification was obtained or an unspecific product was amplified, the absence of a Ct value was replaced by 40, the maximum possible Ct value. In addition, the specificity of PCR products was assessed by melting curve analysis

### 2.8. Mutational Analysis (Whole Exome Sequencing)

After DNA extraction, material was sequenced with 126 bp paired-end reads using the ILLUMINA HiSeq-2500 platform and the Agilent V5 50MB enrichment kit, with a coverage of >160x Next, human reads were isolated using Xenome [16], aligned to the GRCh38 reference genome, and mutations were called using a workflow based on GATK (Genome Analysis Toolkit version 4) best practices. Using the variant effect predictor (VEP) [17], candidate mutations were annotated and filtered considering only variants with moderate or high protein impact and those being rare in healthy populations (<1% in gnomAD) [18]. Furthermore, mutations shared by all three resistant clones were annotated with protein functions using SIFT and Polyphen predictions from SNPnexus [19].

### 2.9. Reverse Phase Protein Array (RPPA)

In total, 150 xenograft samples (30 NSCLC models in five replicates) were provided to Institut Curie, France to perform a Reverse Phase Protein Array (RRPA). The samples were printed onto nitrocellulose covered microscope slides, using an Aushon 2470 precision printer. The samples were prepared as described in [20] and printed onto nitrocellulose covered slides (Supernova, Grace Biolabs) using a dedicated arrayer (2470 arrayer, Aushon Biosystems) in five serial dilutions (2000 to 125 μg/mL) and two replicates per dilution. Arrays were labeled with 33 specific antibodies (see Table 4). Read-out was done using IRDye 800CW (LiCOR) on an Innoscan 710-AL scanner (Innopsys). For the staining of total protein, arrays were incubated 30 min in Super G blocking buffer (Grace Biolabs), rinsed in water, incubated 5 min in 5.10^−6^% Fast green FCF (Sigma), and rinsed again in water. Raw data were normalized using Normacurve [21]. Next, each RPPA slide was median centered and scaled (divided by median absolute deviation). We then corrected for remaining sample loadings effects individually for each array by correcting the dependency of the data for individual arrays on the median value of each sample over all the arrays using a linear regression. Median normalized data were used to compare expression levels between groups of samples. The ratio between phosphorylated and total protein was calculated by calculating the difference between the log-transformed phospho-protein expression and the log-transformed total protein expression.

### 2.10. Protein–Protein Association Network Analysis

The Search Tool for Retrieval of Interacting Genes/Proteins (STRING, https://string-db.org/) was used to assign mutated genes as well as activated proteins to a protein interaction network. The enrichment analysis was based on gene ontology (GO) terms and KEGG pathways [22].

### 2.11. Statistical Analysis

Moderated *t*-tests [23] (followed by Benjamini–Hochberg multiple testing correction) were applied for statistical analysis of RPPA data. For the evaluation of the statistical significance in all other cases, the Mann–Whitney test (two-tailed) was used unless otherwise indicated. By convention, *p*-values ≤ 0.05 indicate significant differences. Statistical calculations were performed using GraphPad Prism bio-analytic software (version 6.02 for Windows, GraphPad Software, San Diego, CA, USA, www.graphpad.com) and R.

## 3. Results

### 3.1. Patient-Derived NSCLC Xenograft LXFA 677 Showed EGFR Dependency and Dose Dependent Sensitivity towards Gefitinib

To identify the best suited PDX model for this study, we screened our NSCLC PDX collection for EGFR protein expression by IHC. We therefore created a TMA comprising all 85 models in duplicates (Figure 1a and Appendix A). To verify these results and to retrieve data for p-EGFR expression, we performed an RPPA with a subset of these models (*n* = 27) in triplicates (Figure 1b). The lung adenocarcinoma model LXFA 677 displayed high expression levels of EGFR and medium levels of p-EGFR as indicated by the median corrected normalized values (Figure 1a,b, black arrow). The EGFR and p-EGFR protein expression of this models was consistently high in both assays. To ensure that not only is the target expressed but is also druggable in vivo, we tested 47 NSCLC PDX towards their sensitivity against nine different EGFR targeting agents. Each model was tested against one to five compounds in total. LXFA 677 displayed a mean optimal T/C value of 23% (Appendix A). In detail, it was sensitive towards treatment with four different EGFR inhibitors, TKIs, and mAb. The multikinase inhibitor, Sorafenib, targeting not exclusively EGFR but also other kinases, such as VEGFR and PDGFR, was only moderately active (Figure 1c). In a more detailed experimental setting, the model showed a dose-dependent sensitivity towards Gefitinib (Figure 1d). Taken together, LXFA 677 was sensitive towards EGFR inhibition, which underlined the essential role of the EGFR pathway, qualifying it as an ideal candidate for the experimental set-up (Figure 1d).

### 3.2. Gefitinib-Resistant Tumors Emerged under Continuous Treatment

To induce acquired resistance, we started treatment with a clinically relevant dose of 50 mg/kg/day and a suboptimal dose of 40 mg/kg/day per oral gavage, respectively (Figure 2a). Once the tumors went into remission, the dosing was reduced to 80% and 50% of the initial concentration. Marked reduction of the tumor volume occurred 7 to 14 days after start of treatment depending on the treatment scheme. When tumors started regrowth, the Gefitinib dosing was adapted accordingly with increasing amounts of compound up to 50 mg/kg/day. The emerging tumors were designated as individual sublines, LXFA 677res 1–3. Although the number of mice per treatment arm was small (two for the high dose and four for the low dose regimen), we found that starting with the higher dose level of 50 mg/kg/day induced resistance more consistently than the second approach, where 80% of the clinically relevant dose was applied to the treatment-naïve tumors. In the latter, only one out of four tumors showed regrowth under Gefitinib treatments whereas two out of two tumors emerged as EGFR inhibition in the first experimental set-up.

When tumors reached a volume of ≥1000 mm^3^, the tissue was harvested and re-implanted in four recipient mice per donor tumor. Treatment started when subcutaneously implanted tumors showed signs of successful engraftment (tumor volume ≥50 mm^3^). The four mice bearing one subline were divided into one treatment arm receiving different doses of Gefitinib and one control arm receiving the control vehicle. All mice allocated to the Gefitinib groups received 60% of the clinically relevant dose (=30 mg/kg/day). Once a tumor volume of >200 mm^3^ was reached, the daily Gefitinib dose was enhanced to 40 mg/kg/day. This treatment regimen was maintained until individual tumors reached at least 1500 mm^3^. Subsequently, tumor tissue was harvested and viably frozen down for further use (Figure 2b). LXFA 677res1 showed similar or even faster growth kinetics in treated tumors compared to untreated tumors. LXFA 677res2 depicted the same tumor growth rate in all mice irrespective of their medication. Both sublines were developed with the high dose induction protocol. In contrast, LXFA 677res3 derived from the low dose induction protocol showed inconsistent tumor growth in the different settings. One tumor from the control arm did not grow at all, whereas the other one depicted a growth kinetic like the two tumors under continuous Gefitinib treatment. The doubling time of the different sublines are shown in Table 5. LXFA 677res1 grows faster than the original tumor models whereas the other lines indicate a similar growth behavior. Due to the small number of animals per group, statistical analysis was not performed. To examine if the acquired resistance was reversible, we transplanted tumor material from one animal of the control arms each into four recipient mice. These mice were then stratified into the control and treatment arm and the tumor volume was monitored over time. Despite the fact that the donor tumor material was without Gefitinib treatment for one passage, the re-implanted tumors showed resistance towards Gefitinib to a similar extent as the original resistant sublines, suggesting that the acquired resistance was not reversible (Appendix A). In a first attempt to decipher the mechanism behind the resistance, the copy number variation and mRNA expression for a subset of genes was determined (Figure 2c). The copy number of cMET was enhanced in all three sublines, whereas the copy number of HER2 was increased in sublines LXFA 677res1 and -res3 and reduced in -res2. Consistently, the mRNA expression levels of all three sublines were enhanced compared to the original model (relative expression to LXFA 677 = 1). The same observation was made for AXL, but to a much lower extent. HER3, HGF, and pTEN were only slightly modulated in the different sublines with a trend towards downregulation.

### 3.3. Whole Exome Sequencing Analysis Revealed High Similarities between the Resistant and the Treatment Naïve Clones of NSCLC PDX LXFA 677

To reveal the possible mechanism(s) of resistance, whole exome sequencing analysis was performed with the three resistant sub-clones of the NSCLC PDX and compared to the respective data set of the original tumor model. In all four lines, no mutations, in particular the T790M mutations, were detected in the *EGFR* gene. Besides that, they shared 85% of their mutated genes (=504) in total (Figure 3a). The total number of mutated genes was also very similar among all lines, displaying only minor differences (Appendix A). A smaller subset of mutations was shared by at least two of the lines or was even unique for one specific subline (= not common; 26 to 42 mutated genes shared by at least two lines; 26 to 53 unique mutated genes). Hierarchical clustering of genes with differing but not unique mutated genes (at least2; n = 53) showed that the original treatment naïve clone clustered separately from the three other lines (Figure 3b). Within the resistant clones, LXFA 677res2 and LXFA 677res3 clustered together. The mutations shared by all three resistant clones (14 mutations in 13 genes) were annotated with protein functions using SIFT and Polyphen predictions from SNPnexus. Functional annotation was available for missense variants (n = 10), whereas it was missing for insertions and a missense variant on the X chromosome (*n* = 4). Of note, seven of these mutations were also identified in an NSCLC PDX from a patient with an acquired resistance towards Erlotinib (LXFE 2478). The missense mutation found in NOMO2, ARHGEF5, and SMTNL2 were predicted as deleterious based on the SIFT prediction. Moreover, NOMO2 was predicted as being possibly damaging by the Polyphen algorithm. Using the STRING platform, we investigated the mutated genes that got acquired by at least two resistant models in comparison to those in the naïve treatment (Appendix A). Tolerating a maximum of five interactors and a medium confidence level (=0.400), the Fanconi anemia pathway (KEGG), cell cycle checkpoints (reactome), and DNA topoisomerase type I activity (GO) showed up as the most important interactors with the lowest false discovery rate and the highest observed gene count (Appendix A).

### 3.4. The NSCLC PDX Lines Emerging under Gefitinib Treatment Displayed Specific Proteomic Profiles

To elucidate how the tumor cells developed a resistance mechanism against Gefitinib, a Reverse Phase Protein Array (RPPA) was performed on the three resistant sublines and the treatment naïve parental clone. Hereby, it was possible to study protein expression and activation (=phosphorylation) in a large panel of samples and proteins in a quantitative manner. The cluster analysis exhibited three distinct groups of proteins: EGFR and AXL showed high expression levels in all four lines. Medium expression levels were determined for phospho−p38 MAPK, phospho−p44/42 MAPK, phospho−PTEN, STAT3, Akt, STAT1, pTEN, NFkB, IGF−I receptor B, MEK1/2, mTOR, RelB, MET, HER3, and phospho−eIF4B. Low expression levels were detected for phospho−mTOR, phospho−Akt, phospho−c−Jun, phospho−STAT1, p44/42 MAPK, p38 MAPK, phospho−Her3, c−Jun, phospho−MEK1/2, eIF4B, IkB alpha, phospho−NFkB, HER2, phospho−EGFR, phospho−STAT3, and phospho−MET (Figure 4a). As the three resistant sublines displayed high similarities, we determined the logarithmic fold change (log FC) by comparing the values of all three resistant lines vs. the original treatment naïve line. Four proteins were upregulated in the treatment resistant lines with a logFC > 0.5. As the sample number was relatively low, *n* = 5 and *n* = 15, respectively, only phospho-STAT1 revealed a statistically significant difference (moderated *t*-test; multiple testing adjusted *p*-value = 0.03). A functional interaction analysis using the STRING platform was performed investigating the interaction of the upregulated proteins (Appendix A). The most prominently involved pathways for the upregulated proteins were the viral process and ERBB2 signaling (GO; Appendix A) and EGFR tyrosine kinase inhibitor resistance as well as prostate cancer (KEGG; Appendix A). The protein data corroborated the transcriptomic results, with cMET being the only exception. While cMET showed enhanced levels of copy numbers as well as an increase in mRNA levels, it did not translate to a higher protein level compared to the original PDX model.

A better understanding of the actual activity of the expressed proteins was achieved by comparing the ratio between phosphorylated and total protein (*n* = 14). The comparison of the ratio between phosphorylated vs. total protein displayed a moderate, albeit not significant, upregulation in the resistant clones for four proteins: MEK1/2, STAT3, STAT1, and NFkB (log FC > 0.5). In contrast, Akt was down-regulated in its activation status (logFC < −0.5, Figure 4b). The STRING platform was used again and potential interaction partners of the up-regulated activated proteins were investigated (Appendix A). The enrichment analysis revealed a highly significant positive correlation with the regulation of signal transduction (GO-term) and EGFR TKI resistance, different solid cancer types, namely pancreatic, prostate, and NSCLC as well as ErbB and HIF-1 signaling (KEGG pathway, Appendix A).

## 4. Discussion

Acquired resistance prevents patients from long term benefits of targeted therapies in many different cancer types. This specifically holds true for anti-EGFR treatment in NSCLC patients, where multiple generations of TKIs all started displaying a loss of activity after a certain period of time [24]. It is therefore even more important to create preclinical models mimicking this phenomenon as close as possible. Studies using tissue culture induced resistance do not take into account that it is not only the tumor cell involved in the development of the resistance mechanism. The complex cross-talk between tumor cells and their micro-environment does not only influence the phenotype of the tumor but its sensitivity towards anticancer agents as well [25]. These aspects can, at least for now, only be mimicked in an animal model. The present study used in vivo PDX of NSCLC to model the onset of resistance against EGFR TKI treatment. A PDX model does recapitulate some aspects of the patient tumor more closely than conventional cell lines, which makes this model an ideal candidate for such a complex task [13].

The PDX model selected in this study represents a typical late stage NSCLC. The fact that the original tumor was EGFR and KRAS wt classified it for second line EGFR treatment [26]. Indeed, it mirrored precisely the clinical situation, with an emerging resistance under constant treatment with Gefitinib.

A preliminary screening for five genes based on the copy number and qPCR led to first conclusions but did not reveal the complete mechanism behind the acquired resistance. Despite the upregulation of cMET on the RNA level, all other gene expression modulations were reflected as well on a protein level. A possible explanation could be the subtle down-regulation of activated eIF4B (logFC < −0.3) being involved in the translation of MET mRNA [27]. Based on the performed whole exome sequence analysis, we identified a set of 13 genes with shared mutations in all three Gefitinib-resistant LXFA 677 subclones (res1 to res3). Interestingly, none of these have yet been directly linked with resistance, except for NOMO2 (NODAL Modulator 2), the mutation which was identified in the current study as prognostically unfavorable by two algorithms (SIFT and Polyphen). NOMO2 together with three other genes has recently linked to radio-resistance and suggested as diagnostic biomarkers in radio-resistant human H460 lung cancer stem-like cells [28]. Functionally, NOMO2 is reported to block Nodal signaling [29]. Furthermore, high levels of Nodal are bad prognostic markers in melanoma, breast, and pancreatic cancer [30]. Thus, it can be hypothesized that the knock-down of the blocking signal from NOMO2 induces an increase of Nodal in the resistant clones as a possible escape mechanism. Of course, this hypothesis maintains to be verified. The two other mutations predicted as deleterious, ARHGEF5 and SMTNL2, are linked to a poor prognosis in lung cancer patients, epithelial-mesenchymal transition (EMT), and Rho-GTPase signaling [31,32]. The biological functions of some identified genes are still not elucidated, which holds true for CCDC74A (coiled-coil domain containing 74A) and OR2T35 (olfactory receptor family 2 subfamily T member 35), for which no publication is available as of Apri 2019. Two other genes, *ZNF417* and *KDM6B*, have been associated as part of a gene signature with cancer aggressiveness [32,33,34,35,36,37,38,39,40]. In a gene enrichment analysis, the Fanconi anemia (FA) pathway was indicated in addition to the above mentioned biological processes. Interestingly, this pathway was reported in connection with EGFR resistance by other groups [41,42,43,44]. Pfaffle et al. explicitly described an FA phenotype in EGFR resistant cells and a possible susceptibility of these cells towards PARP inhibition [43]. This assumption was based inter alia on the work of Garcia-Higuera et al., who described the co-localization of proteins of the FA pathway with BRCA1 [45]. A future study investigating the sensitivity towards PARP inhibitors in the original as well as the resistant clones will help to further characterize this possible link and treatment option.

Although the above discussed genes were found to be mutated in all three Gefitinib resistant clones, it remains to be determined if and how these mutations may contribute to resistance. For most of those genes, only a very vague knowledge about their biologic function in general exists. However, the comparison with the mutational landscape of another NSCLC PDX of our collection, which was derived from a patient with an acquired resistance to Erlotinib, revealed and overlap of 7 out of 14 mutations (in 13 genes) with the current PDX models, where the resistance was acquired in the tumor bearing mice.

The possible trend towards EMT seen in the whole exome sequencing analyses was substantiated by the fact that NFκB activation was enhanced in the resistant clones as determined by RPPA [46]. A biological link between EGFR expression and NFκB activation was described for different solid cancer types, like NSCLC, but also for renal cancer [47,48]. As our PDX model is carrying a p53 mutation, it thereby qualifies as a possible pre-clinical model for the development of NFκB inhibitors. The fact that IκBα was slightly enhanced (logFC > 0.1) in the resistant clones, indicates that the classical NFκB activation pathways seemed to be involved [49]. The activation of EGFR via different upstream proteins, like RET [50] or HER2 [51], was described before. In the present case, the exact activating mechanism of EGFR remains to be elucidated. In any case, when EGFR is overexpressed, it becomes phosphorylated on tyrosine residues located in the C-terminal part and constitutively activated, forming a complex with TBK1 and IRF3 [52]. Additionally, it has been shown that *STAT1* gene expression is upregulated by nuclear EGFR and HER2 via cooperation with STAT3 [53]. This observation is supported by our data. The upregulation of AXL determined in all three resistant lines of LXFA 677 was observed by other groups in vitro as well as in patient samples of NSCLC and other solid cancer types [54,55,56,57]. Tian et al. described the possibility to restore sensitivity towards Gefitinib in NSCLC cell lines by silencing AXL [58]. The observed upregulation of activated MEK1/2 is in line with the comments of other groups that MEK1/2 inhibition can prevent or circumvent EGFR TKI resistance in monotherapy or in combination with PI3K/mTOR inhibitors [59]. The MEK1/2 inhibition prevented the EMT emerging in the EGFR resistant NSCLC cells [60]. As a marked proportion of our data is suggesting EMT as one possible escape mechanism in the current model, an upregulation of MEK1/2 is in line with the published data. In contrast, the activation of STAT1 as well as of STAT3 was up to now, not linked to EGFR resistance. Interestingly, the activation of STAT3 is linked to EMT whereas the role of STAT1 is discussed controversially [61,62,63]. Apart from this, specifically, the upregulation of pSTAT1 under treatment with EGFRi correlated with worse survival in patients undergoing combination therapy with anti-EGFR and Cisplatin [64]. This observation was in line with the trend towards a poor prognosis and aggressiveness of tumor growth elucidated in the whole exome sequencing analysis. Cheng et al. described a correlation between a reduction of PDL-1 under EGFRi treatment via STAT1. The upregulation of activated STAT1 in the acquired resistance model can be interpreted as a possible immune escape mechanism of the tumor cells to upregulate PDL-1 and evade the CD8+ tumor infiltrating lymphocytes [65]. The downregulation of activated Akt has not been described yet for acquired resistance in EGFR TKI treatment. Nevertheless, Akt downregulation via inactivation of Ets1- function was described by two independent groups as an innate drug resistance mechanism in EGFR mutated NSCLC cells [66,67]. The possibility that a similar escape mechanism can evolve under treatment with EGFR TKIs in EGFR wt NSCLC lines needs to be investigated. In order to obtain a more robust understanding about the complex escape mechanisms, it will be of great importance to generate and analyze additional resistant clones with regard to their mutation status, their transcriptional profile, and their phospho-proteomics status.

In summary, the generation of sister tumor cell lines derived from a pre- and post-acquired resistance status makes this PDX panel a valuable tool for the exploration of the biological mechanism behind this phenomenon. The further analysis and utilization of this PDX model may facilitate research and development regarding the prevention and treatment of drug resistance and will certainly contribute towards continued improved treatment and long-term survival in NSCLC patients.

## Figures and Tables

**Figure 1 cells-08-00740-f001:**
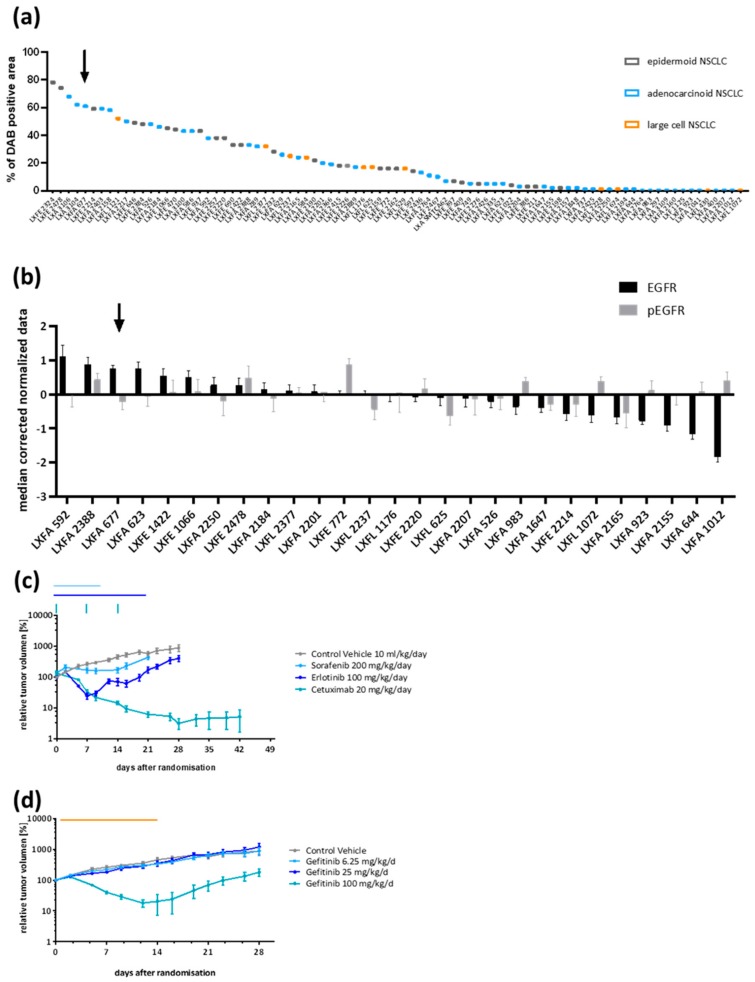
Quantification of EGFR and p-EGFR expression in a panel of NSCLC PDX models. (**a**) EGFR expression was determined by IHC on tissue micro arrays (TMAs) of the NSCLC PDX panel: sections were incubated with anti-human EGFR antibody (1:36) overnight at 4 °C, followed by DAB staining and hematoxylin counterstaining. Digitalized images of the IHC slides were evaluated to determine the percentage of EGFR positive areas using an inhouse software. The computerized analysis was used to quantify the EGFR expression using color classification and morphological image processing techniques. (**b**) The expression of EGFR and p-EGFR was determined by RPPA in a subset of the NSCLC PDX panel (*n* = 27). The samples were printed onto nitrocellulose covered microscope slides in five serial solutions and two replicates per dilution. Arrays were labeled with specific antibodies listed in Table 4. Median normalized data were used to compare expression levels between groups of samples. The ratio between phosphorylated and total protein was calculated by calculating the difference between the log-transformed phospho-protein expression and the log-transformed total protein expression. LXFA 677 showed high levels of EGFR and mean levels of p-EGFR. (**c**) LXFA 677 was implanted subcutaneously in NMRI nude mice and treatment started when a median tumor volume of 250 mm^3^ was achieved. Animals were assigned to the respective treatment arms. Dosing and schedule of the compounds are shown in Table 2. Treatment duration is indicated as lines on top of the figure. LXFA 677 showed EGFR dependent growth due to its sensitivity towards anti-EGFR monoclonal antibody, Cetuximab, as well as selective tyrosine kinase inhibitor (TKI), Erlotinib. Of note, the multi-TKI Sorafenib exhibited only marginal activity. (**d**) LXFA 677 was implanted subcutaneously in NMRI nude mice and treatment started when a median tumor volume of 250 mm^3^ was achieved. Animals were assigned to the respective treatment arms. Dosing and schedule of the compounds are shown in Table 2. Treatment duration is indicated as lines on top of the figure. LXFA 677 displayed a dose-dependent sensitivity towards treatment with Gefitinib.

**Figure 2 cells-08-00740-f002:**
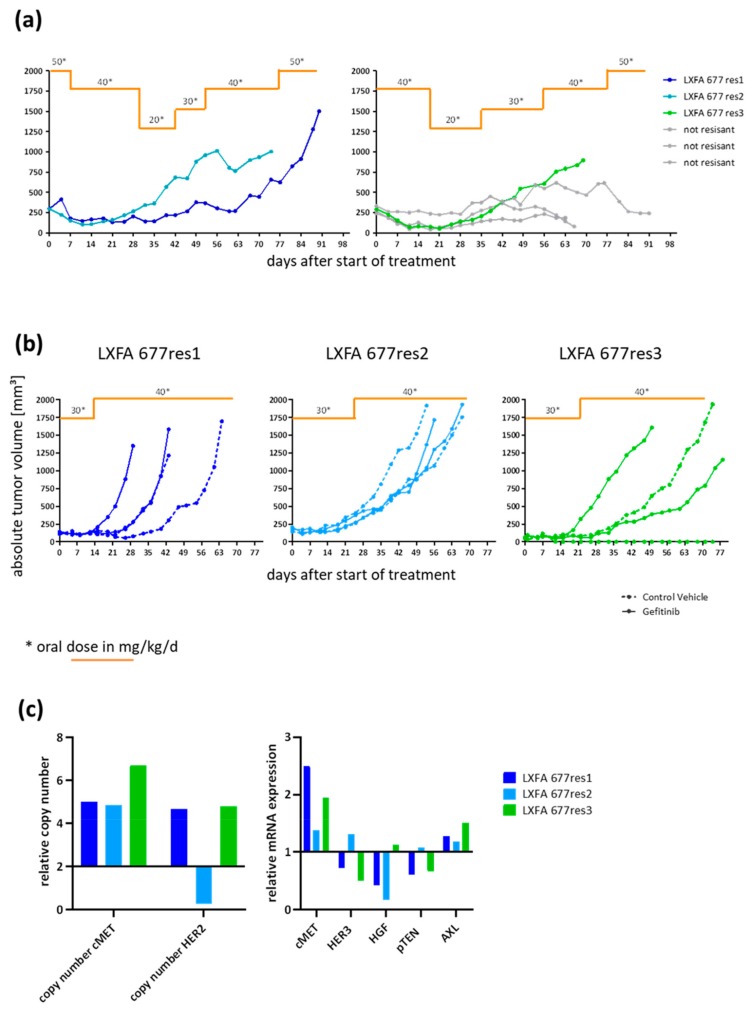
Induction of resistance in NSCLC PDX LXFA 677 in vivo. (**a**): Tumor growth curves for individual tumors under different doses of Gefitinib. The orange line is indicating the duration of the treatment. The respective dose per day is shown above the line. (**b**): Tumor growth curves for individual tumors after re-implantation of tumors emerging under constant Gefitinib treatment. The orange line is indicating the duration of the treatment. The respective dose per day is shown above the line. (**c**): Determination of copy number variation (left diagram) and mRNA expression level (right diagram) was determined for three resistant sublines and the treatment naïve NSCLC PDX LXFA 677. The relative expression was determined by using 2.0 (copy number) and 1.0 (mRNA expression level) as default for the treatment naïve line.

**Figure 3 cells-08-00740-f003:**
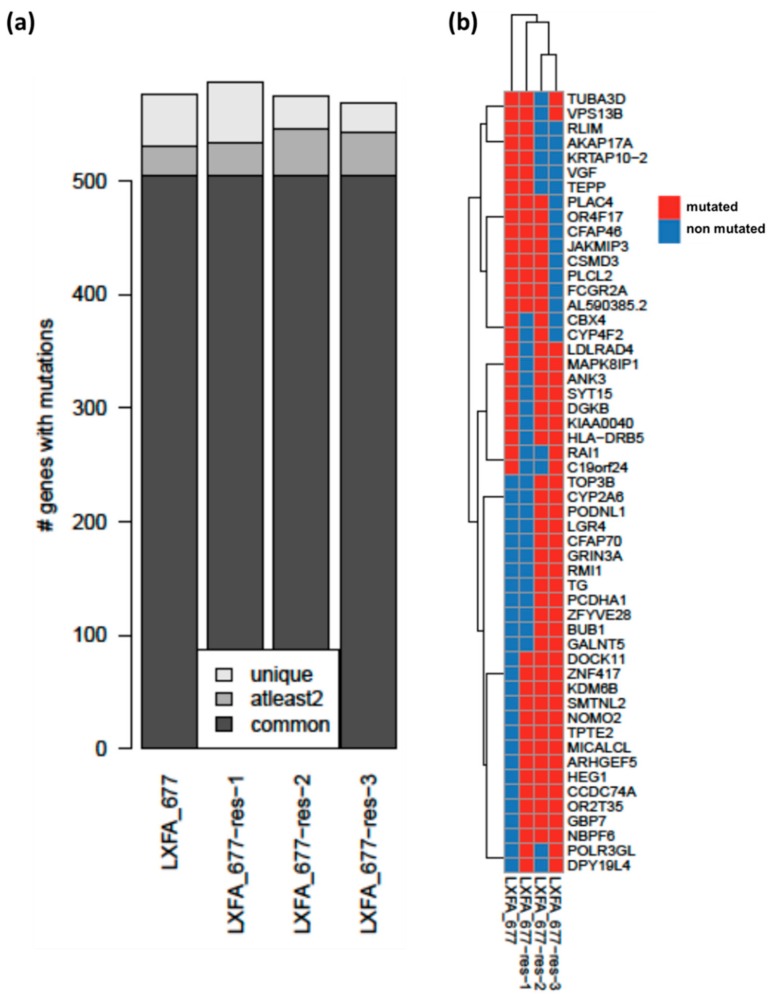
Results of whole exome sequencing analysis of four different LXFA 677 sublines. (**a**): Mutational landscape of the four LXFA 677 sublines with more than 85% (*n* = 504) of genes with mutations being shared across clones. (**b**): Hierarchical clustering of genes with differing (=at least two; *n* = 53) mutation status. The whole exome sequencing was performed using the variant effect predictor (VEP). Candidate mutations were annotated and filtered considering only variants with moderate or high protein impact and those being rare in healthy populations (<1% in gnomAD). Mutations shared by all three resistant clones were annotated with protein functions using SIFT and Polyphen predictions from SNPnexus as depicted in detail in Appendix A.

**Figure 4 cells-08-00740-f004:**
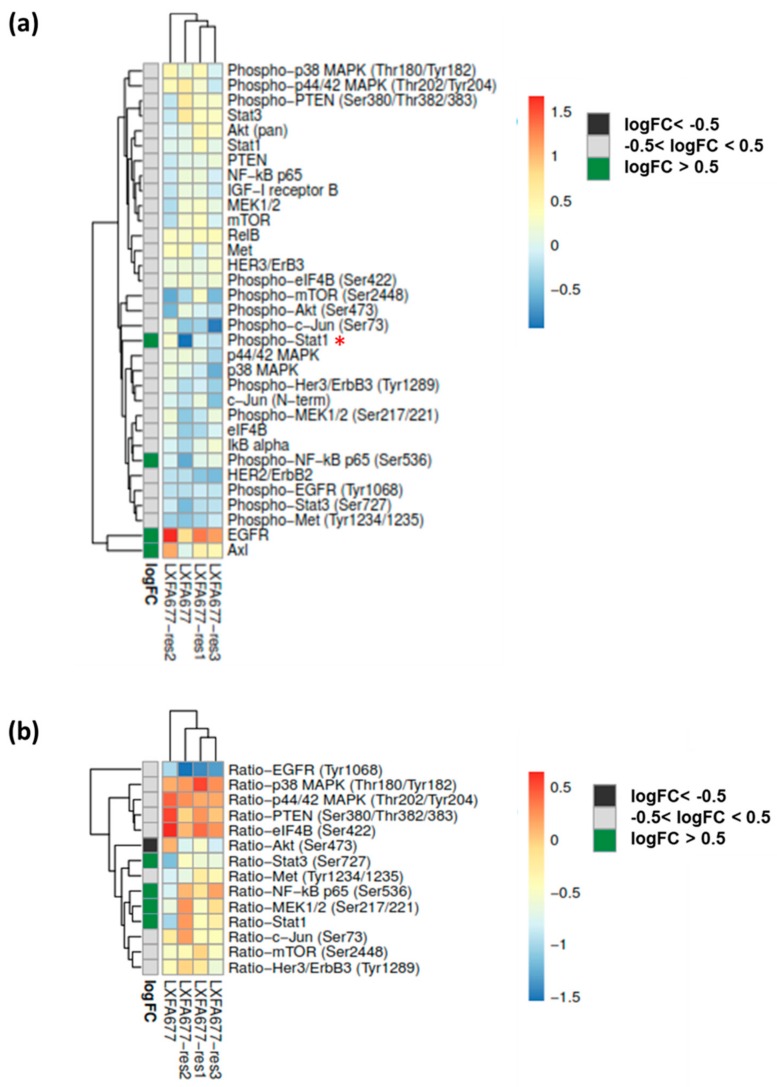
The proteomic landscape was determined by RPPA analysis of four different sublines of LXFA 677. The median corrected normalized values were used for all subsequent data analyses. The logarithmic fold change (logFC) was calculated comparing the sum of all resistant lines with the treatment naïve tumor. (**a**): Hierarchical clustering of average protein expression across five dilutions. (**b**): Hierarchical clustering of average phosphor- to total protein ratios across five dilutions.

**Table 1 cells-08-00740-t001:** Metadata of the donor patient.

Diagnosis	Biopsy	Pre-Treatment	Stage	Pts Age	Pts Gender	Ethnicity	HLA-Type
undifferentiated adenocarcinoma of the lung	primary tumor	Cisplatin	T1N1Mx	62 years	male	Caucasian	A*31:01 B*56:01 DQB1*05:01 DRB1*10:01 DQB1*05:01 DRB1*01:01 A*11:01 B*37:01 C*06:02 no C*01:02 no

**Table 2 cells-08-00740-t002:** Doses and schedules of compounds applied to tumor-bearing animals.

Compound	Supplier	Daily Dose [mg/kg/d]	Route	Schedule
Sorafenib	# S8502, LC Laboratories, USA	200	*per os*	0–11
Lapatinib	#L-4899, LC Laboratories, USA	150	*per os*	0–21
Osimertinib	#O-7200, LC Laboratories, USA	5	*per os*	0–21
Afatinib	#A-8644, LC Laboratories, USA	12.5	*per os*	0–21
Necitumumab	#LY3012211, local pharmacy	20	intraperitoneal	twice weekly for 3 weeks
Erlotinib	# E-4007-25G, LC Laboratories, USA	100	*per os*	0–21
Cetuximab	PZN 00493528, Merck Serono, USA	20	intravenous	0, 7, 14
Sunitinib	#S-8803, LC Laboratories, USA	30	*per os*	0–21
Gefitinib	#G-4408-1G, LC Laboratories, USA	up to 50	*per os*	0—end of experiment

**Table 3 cells-08-00740-t003:** Primers designed for gene copy number and mRNA expression of selected genes.

Target	Forward Primer	Reverse Primer	Used for
**GAPDH**	CAA′ATT′CCA′TGG′CAC′CGT′C	GCC′ACA′CCA′TCC′TAG′TTG′C	Gene copy number (housekeeping gene)
**ERBB2**	CTGAACTGGTGTATGCAGATTGC	TTCCGAGCGGCCAAGTC	Gene copy number
**cMET **	GACATTTCCAGTCCTGCAGTCA	CTCCGATCGCACACATTTGT	Gene copy number/mRNA expression
**ERBB3**	AATAAAAGGGCTATGAGGCGATACT	AGCTTCCTTAGCTCTGTCTCTTTGA	mRNA expression
**PTEN**	CAGTAAGCGTTTTTTTTCTTTGAAGA	TGTGTAAGGTCAAAAGGGTGGAA	mRNA expression
**HGF**	GAATACTGCAGACCAATGTGCT	TTGCAAGTGAATGGAAGTCCT	mRNA expression
**18S **	CTACCACATCCAAGGAAGGCA	TTTTTCGTCACTACCTCCCCG	mRNA expression (housekeeping gene)

**Table 4 cells-08-00740-t004:** Antibodies used for RPPA assay.

Index	Designation	Source	Mono/Polyclonal
1	Phospho-Akt (Ser473) (193H12)	Rabbit	Monoclonal
2	Phospho-PTEN (ser380/Thr382/383)	Rabbit	Polyclonal
3	p44/42 MAPK	Rabbit	Polyclonal
4	Stat3	Rabbit	Polyclonal
5	Phospho-p38 MAPK (Thr180/Tyr182)	Rabbit	Monoclonal
6	Phospho-Stat3 (Ser727)	Rabbit	Polyclonal
7	Stat1	Rabbit	Polyclonal
8	Phospho-MEK1/2 (Ser217/221)	Rabbit	Monoclonal
9	Phospho-p44/42 MAPK (Thr202/Tyr204)	Rabbit	Monoclonal
10	Phospho-c-jun (Ser73) (D47G9)	Rabbit	Monoclonal
11	NF-kB p65	Rabbit	Monoclonal
12	Phospho-NF-kB p65 (Ser536)	Rabbit	Monoclonal
13	Phospho-Met (Tyr1234/1235)	Rabbit	Monoclonal
14	p38 MAPK	Rabbit	Monoclonal
15	phospho-mTOR (Ser2448)	Rabbit	Monoclonal
16	Phospho-Her3/Erbb3 (tyr1289)	Rabbit	Monoclonal
17	HER3/ErbB3 (c-17)	Rabbit	Polyclonal
18	mTOR	Rabbit	Polyclonal
19	c-Jun (N-term)	Rabbit	Monoclonal
20	Phospho-eIF4B (Ser422)	Rabbit	Polyclonal
21	eIF4B	Rabbit	Monoclonal
22	MEK1/2	Rabbit	Polyclonal
23	Akt (pan) (C67E7)	Rabbit	Monoclonal
24	PTEN (D4.3) XP	Rabbit	Monoclonal
25	Phospho-Stat1 (Y701)	Rabbit	Monoclonal
26	EGFR (D38B1)	Rabbit	Monoclonal
27	RelB	Rabbit	Monoclonal
28	Met (D1C2) XP	Rabbit	Monoclonal
29	Axl (C2B12)	Rabbit	Monoclonal
30	IkB alpha (44D4)	Rabbit	Monoclonal
31	HER2/ErbB2	Mouse	Monoclonal
32	IGF-I receptor B	Rabbit	Monoclonal
33	Phospho-EGFR (Tyr1068) (D7A5)	Rabbit	Monoclonal

**Table 5 cells-08-00740-t005:** Doubling times of different sublines of NSCLC PDX LXFA 677.

	LXFA 677 Untreated	LXFA 677 Treated	LXFA 677res1 Untreated	LXFA 677res1 Treated	LXFA 677res2 Untreated	LXFA 677res2 Treated	LXFA 677res3 Untreated	LXFA 677res3 Treated
Mean [d]	11.85	22.98	7.54	5.37	15.03	14.43	13.68	14.38
Std. Deviation [d]	0.07	7.62	0.86	0.23	2.36	3.78	0.00	2.31
Nr. of values [n]	2	2	2	2	2	2	1	2

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
