# Peer review of "Induction of Acquired Resistance towards EGFR Inhibitor Gefitinib in a Patient-Derived Xenograft Model of Non-Small Cell Lung Cancer and Subsequent Molecular Characterization"

_cells, 2019, doi:10.3390/cells8070740_

Round 1
Reviewer 1 Report
In this article, authors describe resistance landscapes to the EGFR-TKI using in-house PDX model. Author derived the resistance in mouse model upon long-term treatment of EGFR-TKI sensitive tumor cells. To delineate the mechanism of resistance they use Whole exome sequencing and RPPA. This study reveals the role of Axl, NFkB, Stat1/Stat3 and found mutations in NOMO2, ARHGEF5 and SMTNL2. Authors claim that this could be a preclinical tool to identify further resistance mechanisms to the scientific community.
Importance of this study:
Acquired resistance is the main bottleneck for the success of EGFR-TKI. Hence making the EGFR-TKI derived sensitive and acquired resistance model is of good use to the scientific community. However, in this study authors are using Gefitinib to derive the resistance. With the recent approval of Osimertinib as first-line therapy, gefitinib offers little value to the scientific community.
Major comments
1. Introduction: List out the resistance mechanism to current EGFR-TKI: include the novel EGFR mutations, as EGFR-independent Resistance mechanisms
2. Methods: It is unclear how many resistance models are made, is this only form 1 patient. Please include the statistics on how many patients were used, how many mice were treated to derive resistance to EGFR-TKI
3. Methods: How long does it take for the resistance to develop and was it uniform across the treated animals?
4. T790M is a common mutation resistance mechanism to 1st gen EGFR-TKI: Did they observe this?
5. Was resistance reversible: Sharma et al. describe the reversibility of resistance upon drug withdrawal. Authors should examine this to discuss pre-existing genetic vs non-genetic resistance mechanism.
6. Non-genetic resistance mechanism: Currently NFkB, HDAC and Aurora kinase activation is described as a resistance mechanism to EGFR-TKI and authors should perform the validation on their models to demonstrate that model enriches for the known resistance mechanism as well.
7. Functional consequences of NOMO2, ARHGEF5, and SMTNL2 mutations: examine the downstream effect.
8. The author should compare the LXFA_677 derived acquired resistant tumors vs PDX made from Gefitinib resistance patients: Do you see the same mechanisms?
Reviewer 2 Report
Using patient derived xenograft (PDX) model to mimic the original induction of EGFR resistance in patients, the authors bring forward a more reliable technique to answer this very important question of acquired resistance to drugs under treatment in cancer patients. They also tried to answer the underlying molecular mechanism by using some advanced protein analysis techniques along with bioinformatics based prediction. However, several points which needs attention are as follows.
Major:
1. In abstract, although the authors described the possible underlying molecular mechanism to Gefitinib (EGFR inhibitor) resistance- such as upregulation or downregulation of some genes, but in the summary of their abstract, they only focused on the development of PDX model to study drug resistant. This may need some attention.
2. DAB positive area (indicative of EGFR expression level) of LXFA 2388 had moderate value in Figure 1 (a) where in Figure 1(b), it also showed high level of both EGFR and p-EGFR expression. Anyway, the author selected LXFA 677 stating that it has both high expression level of EGFR and p-EGFR. More discussion on this might be necessary to claim LXFA 677 as the potential one to run further experiments. In addition, the figure legends should have some more information on experimental condition.
3. In Figure 2, the author stated that they used two different strategies to observe the emergence of Gefitinib-resistance. But, the number of mice used in both strategy (one is high dose regimen and the other is low dose regimen) was different- the reason and rationale of which needs to be answered in detail.
4. In Figure 2 (b), why 50 mg/ kg/ day dose was not used in in their experimental setting one where their data presents higher tumor volume in LXFA 677 res 1 at 50 mg/ kg/ day dose?
5. Why in Figure 2 (b) left panel (LXFA 677 res3) the tumor growth of the control is dramatically lower than others in LXFA 677 res2 or LXFA 677 res1?
6. In Figure 2 (c), why the authors did not check HER2 mRNA level as they found its increase in copy number in at least 2 resistant model?
7. The authors mentioned about the underlying molecular mechanism of the acquired Gefitinib resistance in discussion section where it was also hypothesized that this resistance might induce EMT as NFκB was activated in their whole exome sequencing data. Is there any data supporting this claim other than NFκB activation in whole exome sequencing?
8. The summary (in discussion section) was only focused on the implying PDX model to study drug resistance lacking their possible mechanism in brief as their study title not only refers to the development of PDX model but also the underlying molecular mechanism.
Minor:
1. The figure legends should contain more information on the experimental condition and the data. Some results are derived from the bioinformatics analysis using prediction software such as SIFT, Polyphen which are not written in figure legends.
2. The author might add some more information on the molecular mechanism of EGFR resistant based on already published studies in introduction.
3. In Figure 1, the figure legends should have the elaboration or information about DAB.
4. The author should put the information about the X-axis of their data in Figure 2 (a).
5. The experimental design is good enough. However, the explanation of the whole process should be improved.
6. I believe that patient-derived xenograft models of non-small cell lung cancer are the main point of this study that needs more definition about the methods.
7. Please give more information about LXFA 677 in the introduction part and please describe how to modulate LXFA 677 gefitinib-resistant sublines.
8. How could it be generated LXFA 677 gefitinib-resistant sublines for the further experiments? Please discuss it.
9. The Authors give us EGFR and p-EGFR expression level in NSCLC PDX models. First of all: I would like to see the picture of tissue microarray (TMA) and immune-histochemistry about the expression level of EGFR and p-EGFR (if already shown in supplementary figures, please ignore this part).
Second: Authors also should give as the definition of the differences between LXFA 677 and other LXFA sublines.
10. What are the differences between LXFA 677 resistant 1, 2, and 3? Please discuss.
Round 2
Reviewer 1 Report
I am happy with the revision and authors to attempt to provide a feedback